# RUBRICS AS REWARDS: REINFORCEMENT LEARNING BEYOND VERIFIABLE DOMAINS

**Anisha Gunjal**  **Anthony Wang**[*]  **Elaine Lau**[*]  **Vaskar Nath**[*]  **Yunzhong He**
**Bing Liu**  **Sean Hendryx**[*]
Scale AI
anisha.gunjal@scale.com

## ABSTRACT

Reinforcement Learning with Verifiable Rewards (RLVR) has proven effective for complex reasoning tasks with clear correctness signals such as math and coding. However, extending it to real-world reasoning tasks is challenging, as evaluation depends on nuanced, multi-criteria judgments rather than binary correctness. Instance-specific rubrics have recently been used in evaluation benchmarks to capture such judgments, but their potential as reward signals for on-policy post-training remains underexplored. We introduce **Rubrics as Rewards (*RaR*)**, an on-policy reinforcement learning method that extends RLVR beyond verifiable domains by using rubric-based feedback. Across both medical and science domains, we evaluate multiple strategies for aggregating rubric feedback into rewards. The best RaR variant achieves relative improvements of up to 31% on HealthBench and 7% on GPQA-Diamond over popular LLM-as-judge baselines that rely on direct Likert-based rewards. These results demonstrate that RaR-trained policies adapt well to diverse evaluation formats, performing strongly on both rubric-based and multiple-choice tasks. Moreover, we find that using rubrics as structured reward signals yields better alignment for smaller judges and reduces performance variance across judge scales.

## 1 INTRODUCTION

Reinforcement Learning with Verifiable Rewards (RLVR) has enabled large language models to elicit complex reasoning on tasks with clear verifiable outcomes. This is especially effective in domains like math and code, where reward models can be replaced by scoring functions or test cases that automatically verify correctness (Lambert et al., 2024; Guo et al., 2025a; Cui et al., 2025). However, extending RLVR to unstructured, real-world reasoning is challenging because such tasks lack easily verifiable answers. A common workaround is to use preference-based reward models, but they tend to overfit superficial artifacts (e.g. response length, formatting quirks, annotator biases) (Singhal et al., 2023; Wang et al., 2024a; Chen et al., 2024b; Ye et al., 2024; Gudibande et al., 2023) and require large volumes of pairwise comparisons (Ouyang et al., 2022). Instance-specific rubrics have recently emerged for nuanced evaluation in expert domains (Arora et al., 2025), yet their application in on-policy training for expert-level reasoning is largely unexplored.

To address this gap, we explore a paradigm shift that introduces a middle ground between the simplicity of verifiable rewards and the expressiveness of preference rankings, which often come with human artifacts and operational overhead. We introduce **Rubrics as Rewards (RaR)**, a framework for on-policy Reinforcement Learning that uses structured criteria or *rubrics* as the core reward mechanism. Rather than using rubrics only for evaluation (Arora et al., 2025; Sirdeshmukh et al., 2025), we treat them as checklist-style supervision that produces reward signals for on-policy RL. Each rubric is composed of modular, interpretable subgoals that provide automatable feedback aligned with expert intent. By decomposing "what makes a good response" into tangible, human-interpretable criteria, rubrics offer a middle ground between binary correctness signals and coarse preference rankings.

---

[1]Work conducted while at Scale AI.

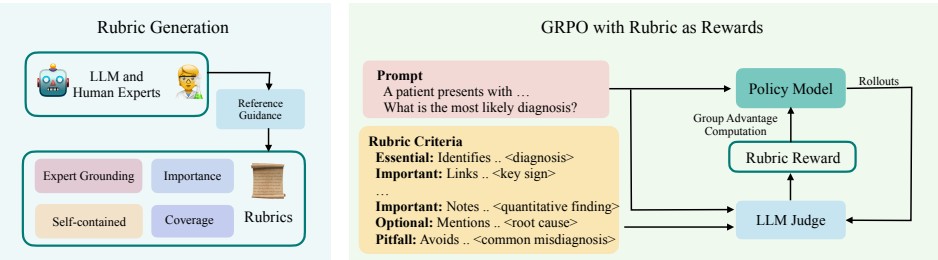

Figure 1: Overview of Rubrics as Rewards (RaR). **(i) Rubric Generation:** We synthesize prompt-specific, self-contained rubric criteria using a strong LLM guided by four core design principles, with reference answers serving as proxies for expert supervision. **(ii) GRPO Training:** These rubrics are used to prompt an LLM judge for reward estimation, which drives policy optimization via the GRPO on-policy learning loop.

Previous works train generative reward models that learn to evaluate reasoning or final outputs with interpretable scores (Chen et al., 2025; Whitehouse et al., 2025; Anugraha et al., 2025; Guo et al., 2025b), and some have even used a model's internal confidence estimates as a proxy for reward (Zhao et al., 2025). More recent efforts have extended verifiable datasets beyond STEM domains, broadening the applicability of RLVR methods to a wider range of tasks (Su et al., 2025b; Ma et al., 2025). Yet a general-purpose approach for specifying reliable reward signals remains elusive, particularly in tasks without a single ground truth where both subjective and objective criteria must be considered. In contrast, we treat rubrics as instance-specific, reusable reward functions. Once generated, rubrics provide interpretable and automatable supervision that can be applied consistently across new rollouts, offering a scalable and transparent alternative to opaque reward modeling in on-policy learning.

Recent concurrent works explore checklists and principled rubric criteria for preference tuning and LLM safety (Gallego, 2025; Viswanathan et al., 2025; Dineen et al., 2025), highlighting a growing trend toward structured supervision. In contrast, we convert rubrics into reward functions for on-policy RL, targeting expert reasoning and applied real-world domains. This closes the rubric-to-learning loop and improves performance on both rubric-guided evaluations and tasks with verifiable answers. Figure 1 illustrates our framework.

Our key contributions are as follows: (i) We introduce **Rubrics as Rewards (RaR)**, an on-policy reinforcement learning framework that uses checklist-style rubrics for multi-criteria supervision in reasoning and real-world domains. (ii) We synthesize instance-specific rubrics for medicine and science and release the corresponding training sets, *RaR-Medicine* and *RaR-Science*. [1] (iii) *RaR*-trained models consistently outperform strong baselines and yield a stable, generalizable training signal, with gains on both rubric-scored and verifiable multiple-choice evaluation settings. (iv) Our results demonstrate that rubric-based rewards provide stable supervision across judge sizes, helping smaller models align effectively with human preferences and maintaining robust evaluation performance from small to large judges.

## 2 RUBRICS AS REWARDS

### 2.1 PROBLEM FORMULATION

Let $x$ denote an input prompt and $\hat{y} \sim \pi_\theta(\cdot \mid x)$ be a sampled response from a model parameterized by $\theta$. In domains without single ground-truth answers or automatic correctness signals, we define a structured reward function using instance-specific rubric criteria.

---

[1]Datasets released at `https://huggingface.co/collections/ScaleAI/rar`.

Each prompt $x$ is associated with a set of $k$ rubric items $\{(w_j, c_j)\}_{j=1}^k$, where $w_j \in \mathbb{R}$ denotes the weight of criterion $j$, and $c_j : (x, \hat{y}) \mapsto \{0, 1\}$ is a binary correctness function that indicates whether the response $\hat{y}$ satisfies that criterion given the prompt.

## 2.2 REWARD AGGREGATION STRATEGIES

We investigate two complementary approaches for combining rubric feedback into scalar rewards:

**Explicit Aggregation.** Each criterion is independently evaluated using an LLM-as-judge, and the final normalized reward is computed as:

$$r(x, \hat{y}) = \frac{\sum_{j=1}^k w_j \cdot c_j(x, \hat{y})}{\sum_{j=1}^k w_j} \tag{1}$$

Normalization makes rewards comparable across prompts that differ in rubric count or weights. Although we use binary checks for $c_j$ in our experiments, the formulation can be extended to continuous-valued scores.

**Implicit Aggregation.** All rubric criteria along with categorical weights are passed to an LLM-as-judge, delegating the aggregation to the model itself to produce a single scalar reward:

$$r_{\text{implicit}}(x, \hat{y}) = f_\phi(x, \hat{y}, \{d_j\}_{j=1}^k) \tag{2}$$

Here, $f_\phi$ denotes an LLM-based judge that takes the prompt $x$, the response $\hat{y}$, and the set of rubric criteria $\{d_j\}$ as input. This formulation allows the model to compute a holistic reward score directly, avoiding the need to manually tune rubric weights.

The prompts used for each method are detailed in Appendix A.7.

## 2.3 GENERALIZATION OF RLVR WITH RUBRICS AS REWARDS

Rubric-based reinforcement learning extends the standard RLVR (Reinforcement Learning with Verifiable Rewards) setting by supporting multi-dimensional, prompt-specific evaluation criteria. We formalize this relationship below.

**Remark 1** (**Rubrics as Rewards subsumes RLVR**). The RLVR setting is a special case of rubric-based rewards defined in Equation 1, where $k = 1$, $w_1 = 1$, and $c_1(x, \hat{y})$ reduces to a single verifiable correctness function that compares the model output $\hat{y}$ against the known correct answer $y$. For example, this could involve exact match or test case execution. Formally:

$$r_{\text{RLVR}}(x, \hat{y}) = \text{match}(y, \hat{y}) \tag{3}$$

where $\text{match}(y, \hat{y}) \in \{0, 1\}$ indicates whether the response satisfies the verifiable correctness condition.

Rubric-based reward functions thus generalize RLVR by enabling multi-dimensional supervision, flexible weighting across criteria, and the incorporation of both objective and subjective aspects of response quality. This formalization highlights that RLVR can be seen as a restricted instance of rubric-guided RL with a single essential criterion. In contrast, rubric-based rewards further enable structured supervision in settings where correctness is multifaceted and may not be strictly verifiable.

## 3 RUBRIC GENERATION

### 3.1 DESIDERATA

A rubric specifies criteria for high-quality responses and provides human-interpretable supervision. We identify four desiderata for effective rubric generation:

**Grounded in Expert Guidance.** Rubrics should reflect domain expertise by capturing the essential facts, reasoning steps, and conclusions necessary for correctness. Ideally, this grounding comes from human experts or their high-quality proxies.

**Comprehensive Coverage.** Rubrics should span multiple dimensions of response quality, including factual accuracy, logical coherence, completeness, style, and safety. Negative criteria (*pitfalls*) help identify frequent or high-risk errors that undermine overall quality.

**Criterion Importance.** Rubrics should reflect that some dimensions of response quality are more critical than others. For example, factual correctness must outweigh secondary aspects such as stylistic clarity. Assigning weights to criteria ensures this prioritization, whether through simple categorical tags, explicit numeric values, or learned weighting schemes.

**Self-Contained Evaluation.** Each rubric item should be independently actionable, allowing either human annotators or automated judges to assess it in isolation without requiring external context or domain-specific knowledge.

### 3.2 RUBRICS CREATION

We apply these desiderata to datasets for reasoning tasks in medicine and science. Given the scarcity of human-annotated rubric datasets in these domains, we use LLMs to generate instance-specific rubrics from golden reference answers at scale, enabling the study of structured rewards without costly human annotation.

For each prompt, an LLM generates a rubric of 7–20 self-contained items. Each item is assigned both a numeric and a categorical weight reflecting its relative importance. While numeric weights provide fine-grained prioritization, in our experiments we adopt categorical labels (*Essential*, *Important*, *Optional*, *Pitfall*) for ease of implementation and interpretability in controlled settings. The resulting rubrics are then used directly as reward functions through either explicit aggregation (Eq. 1) or implicit aggregation (Sec. 2.2).

In practice, we generate rubrics using OpenAI's `o3-mini` and `GPT-4o` (OpenAI o3-mini, 2025; Jaech et al., 2024; Hurst et al., 2024), conditioning generation on reference answers from the underlying datasets to approximate expert grounding. The resulting collections—*RaR-Medicine* and *RaR-Science*—are released publicly. These rubric sets supervise policy models with GRPO using both explicit and implicit reward aggregation.

## 4 EXPERIMENTS

### 4.1 DATASETS

We investigate the utility of rubrics as rewards across two reasoning domains, *medicine* and *science*.

- **RaR-Medicine:** A dataset of 20k prompts drawn from diverse medical reasoning sources, including `medical-o1-reasoning-SFT` (Chen et al., 2024a), `natural_reasoning` (Yuan et al., 2025), SCP-116K (Lu et al., 2025), and GeneralThought-430K (General Reasoning, 2025). Instance-specific rubrics for this dataset are generated with `GPT-4o` (see Appendix A.2).
- **RaR-Science:** A dataset of ∼20k prompts curated to align with GPQA-Diamond categories. Prompts are sourced from `natural_reasoning` (Yuan et al., 2025), SCP-116K (Lu et al., 2025), and GeneralThought-430K (General Reasoning, 2025), covering a broad range of scientific reasoning tasks (Appendix A.3). Rubrics for this dataset are synthesized with `o3-mini`.

### 4.2 TRAINING DETAILS

We conduct all experiments using on-policy reinforcement learning with the GRPO algorithm (Shao et al., 2024), taking `Qwen2.5-7B` as the base policy. Models are trained with a batch size of 96, a learning rate of $5 \times 10^{-6}$, and a constant schedule with 10% linear warmup. Complete hyperparameter settings are listed in Appendix A.4. Training runs are executed on a single compute node equipped with 8 NVIDIA H100 GPUs.

Our training pipeline consists of the following key components:

**Response Generation:** For each prompt $q$, we sample $k = 16$ responses from the current policy $\pi_\theta$, using a context length of 3584 and a sampling temperature of 1.0.

**Reward Computation with Rubrics:** We use `gpt-4o-mini` as the judge model to assign rewards $R_q$ to the sampled responses. We experiment with various reward computation and aggregations strategies further described in Sections 4.3 and 4.4.

**Policy Update:** The policy weights are updated using GRPO based on the computed rewards.

| Category | Method/Baseline | Trained | Aggregation Type | Reward Grounding |
|---|---|---|---|---|
| Rubric-Free | OFF-THE-SHELF | × | N/A | N/A |
| Rubric-Free | DIRECT-LIKERT | ✓ | Single Likert score | N/A |
| Rubric-Free | REFERENCE-LIKERT | ✓ | Single Likert score | Reference answer |
| Rubric-Based | RaR-PREDEFINED | ✓ | Explicit aggregation | Instance-agnostic rubrics |
| Rubric-Based | RaR-EXPLICIT | ✓ | Explicit aggregation | Instance-specific rubrics |
| Rubric-Based | RaR-IMPLICIT | ✓ | Implicit aggregation | Instance-specific rubrics |

Table 1: Overview of rubric-free and rubric-based methods and baselines.

## 4.3 RUBRIC-FREE BASELINES

We consider various rubric-free baselines and off-the-shelf post-trained models. Rubric-free baselines are trained with `Qwen2.5-7B` as the base policy.

**OFF-THE-SHELF:** For off-the-shelf baselines we evaluate performance on `Qwen2.5-7B`. We also include the performance of `Qwen2.5-7B-Instruct` to compare with instruction-tuned variant of the base policy.

**DIRECT-LIKERT:** An LLM-as-judge provides a direct assessment for each response–prompt pair on a 1–10 Likert scale (Zheng et al., 2023; Kim et al., 2024), normalized to $[0, 1]$. The resulting score is used directly as the reward signal for training.

**REFERENCE-LIKERT:** An LLM-as-judge compares the generated response against a reference answer (written by experts or stronger LLMs) and assigns a 1–10 Likert score (Zheng et al., 2023), normalized to $[0, 1]$. This reference-guided score is used as the reward signal for policy updates. The reward for each *(prompt, response, reference)* triplet is defined as:

$$R_{\text{ref}}(q, x) = \text{Norm}(\text{LikertScore}(q, x, x^*))$$

where $x^*$ denotes the reference answer.

## 4.4 RUBRIC-GUIDED METHODS

**RaR-PREDEFINED:** This method uses a fixed set of generic rubrics for all prompts (e.g. *response is concise, response contains correct information*). It employs the Explicit Aggregation method (Equation 1) with all criteria weighted uniformly (see Appendix A.6).

**RaR-EXPLICIT:** This variant also uses Explicit Aggregation using a weighted sum (Equation 1) but applies it to instance-specific rubrics from Section 3. We manually assign numerical weights based on the generated categorical labels: $\{"Essential": 1.0, "Important": 0.7, "Optional": 0.3, "Pitfall": 0.9\}$[2].

**RaR-IMPLICIT:** This variant uses the Implicit Aggregation method (Equation 2). It leverages prompt-specific rubrics, where a judge model evaluates the response as a whole to assign a single Likert rating (1–10), avoiding the need for hand-tuned weights. The reward is normalized to the $[0, 1]$ range during training.

An overview of all baselines and methods is summarized in Table 1.

## 4.5 EVALUATION SETUP

**Rubric-Based Evaluation** We evaluate models trained with *RaR-Medicine* on Health-Bench (Arora et al., 2025), a benchmark of 5,000 clinical conversations designed to assess model

---

[2]Pitfall criteria are phrased in positive form (e.g., "The response avoids misinformation"), so satisfying them contributes positively to the score. If a pitfall is not satisfied, the corresponding reward is reduced or penalized.

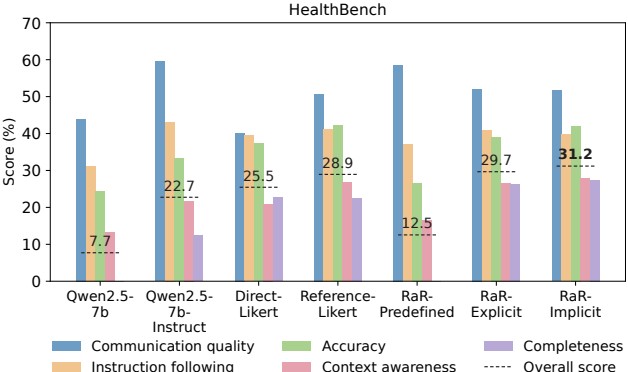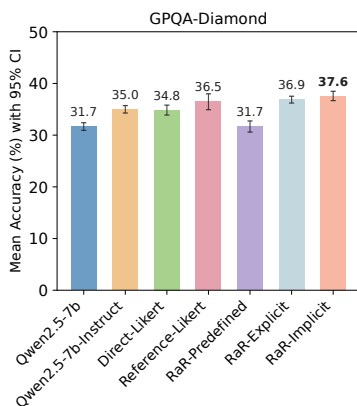

Figure 2: Performance of baselines and RaR (*Rubrics as Rewards*) variants for the medicine and science domains. **HealthBench (left):** shows per-axis scores across five core axes, with a thin dashed gray line indicating the overall score (all values shown as percentages). **GPQA-Diamond (right):** mean accuracy over 10 runs; error bars represent 95% confidence intervals. All policies are evaluated using `gpt-4o-mini` as the LLM-as-Judge. Across both domains, `RaR-Implicit` consistently outperforms `Direct-Likert` and demonstrates a competitive advantage over `Reference-Likert`.

safety and helpfulness in realistic medical scenarios. Performance is measured using detailed, physician-authored rubrics. We generate responses with greedy decoding (temperature = 0) and report both overall scores and per-axis scores following the original setup. For ablation studies, we sample a subset of 1,000 prompts (hereafter referred as HealthBench-1k) and use the rest for training.

**Multiple-Choice Evaluation** Each model is evaluated across 10 independent runs, using greedy decoding (*temperature=0*) to sample one response per prompt. Answer choices are permuted per example to reduce positional bias, and outputs are parsed for boxed answer formats (e.g., `boxed{A}`). If extraction fails, we fall back to a GPT-4o verifier that checks whether the response contains the correct option letter or text (see Appendix A.5). Final accuracy is reported as the mean over 10 runs, and we include 95% confidence intervals to account for run-to-run variance.

**LLM-Judge Alignment Evaluation** To measure how well LLM judges align with human preferences, we build a paired evaluation set from roughly 3,000 HealthBench prompts. For each prompt, we take the practitioner-approved answer as the *preferred* response and create a *perturbed* alternative via controlled edits (see Appendix A.10 for method used for perturbation and prompt selection). The metric is *pairwise preference accuracy* i.e. the fraction of pairs where the preferred response scores higher reported across judge models of varying sizes.

## 5 RESULTS

We now present the main findings of our study.

**Rubrics as Rewards shows strong gains across evaluation settings.** Table 2 reports results on HealthBench (rubric-based, free-form) and GPQA-Diamond (multiple-choice). `RaR-Implicit` consistently outperforms `Direct-Likert`, with relative gains up to 31% on HealthBench and 7% on GPQA. Both rubric-guided variants achieve higher scores than the base and instruction-tuned policies. Gains on GPQA-Diamond show that rubric-induced skills generalize beyond rubric-based evaluation. The `RaR-Predefined` variant, which applies a fixed list of generic rubrics to every prompt (no instance-specific synthesis), underperforms because generic criteria miss prompt-specific requirements and common failure modes, producing misaligned reward signals. Hence, effective training requires instance-specific rubric synthesis as they better capture task context and typical failure modes.

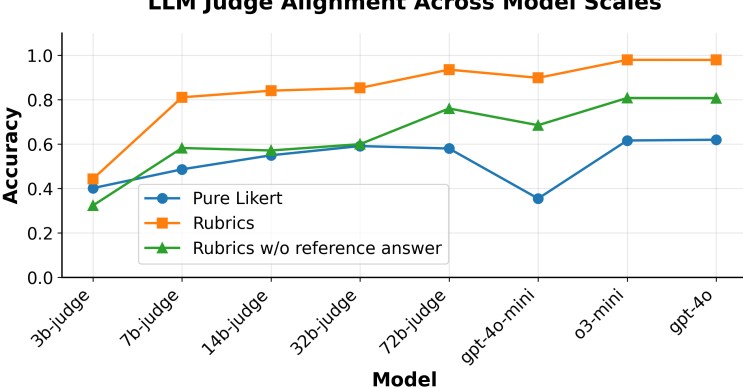

Figure 3: **Alignment Study of LLM Judges across Model Scales.** Rubrics as Rewards (orange) consistently improves alignment with human preferences across LLM judge sizes compared to direct Likert-based scoring (blue). Judge Alignment using synthetic rubrics without expert grounding (green) outperform the direct Likert baseline, but still fall short of expert-grounded rubrics (orange). The rubric structure especially benefits smaller judge models, helping them close the gap with larger models when guided by checklist-style criteria.

Beyond these gains, `RaR-Implicit` also shows small but consistent gains over `Reference-Likert`. In our setup, rubrics are generated with stronger LLMs using reference answers as proxies for expert supervision, so rubric quality is impacted by reference quality. Even so, converting open-ended answers into explicit criteria yields effective, well-aligned reward signals.

Between the two rubric-guided methods, `RaR-Implicit` attains the strongest results overall; fixed weighted sums in `RaR-Explicit` offer more control but can be brittle. Explicit weighting can be difficult to tune but offers greater interpretability; we view the choice as application-dependent and leave it to practitioners. Future work could explore learned or dynamic weighting strategies that maintain interpretability while improving adaptability.

**Rubrics enhance alignment with human preferences across model scales** We evaluate alignment with humans by having LLM judges of varying sizes score chosen vs. rejected HealthBench-1k responses on a 1–10 scale under two settings: (i) rubric-guided (`RaR-IMPLICIT`), where the instance-specific rubric is provided, and (ii) rubric-free (`DIRECT-LIKERT`), where only the prompt and answers are shown. Figure 3 reports *pairwise preference accuracy* (the fraction of pairs where the preferred response receives the higher score). Rubric guidance improves accuracy for every judge size, with the largest gains for smaller judges, narrowing the gap to larger models. This indicates that explicit, context-specific criteria help judges distinguish subtle quality differences better than direct Likert scoring. Further analysis of judge-scale effects on GRPO training is detailed in Appendix A.9.

**Expert guidance is crucial for synthetic rubric generation** Human guidance significantly influences the effectiveness of rubrics in capturing subtle human preferences. Figure 3 highlights performance differences between rubric-based evaluations that include reference answers and those without them. The data shows that rubrics developed with reference answers achieve higher accuracy, emphasizing that human insights integrated during rubric generation enable granular criteria and improved alignment with human preferences.

## 6 ABLATIONS AND ADDITIONAL ANALYSES

**Impact of Rubric Generation Strategies in Real-World Domains** How does the method of rubric generation affect downstream training in challenging, real-world settings? To investigate this, we hold out HealthBench-1k for evaluation and use 3.5k prompts from the remaining Health-Bench pool to generate rubrics for training as it has access to human generated rubrics. Results are summarized in Table 2.

| Training Method | Overall Score |
|---|---|
| Expert-Answer-SFT | 20.4% |
| Simple-Likert | 23.9% |
| Reference-Likert | 31.7% |
| RaR-Implicit-Synthetic-NoRef | 32.0% |
| RaR-Implicit-Synthetic | 35.9% |
| RaR-Implicit-Human | 34.8% |

Table 2: Evaluation on HealthBench: Comparison of human- vs. synthetic-generated rubrics (with and without reference answers). RaR methods trained with GRPO significantly outperform Likert-only, Reference-based-likert and SFT baselines. Synthetic rubrics generated *without* access to reference answers perform notably worse, highlighting the importance of human-grounded guidance. Notably, human-authored rubrics and synthetic rubrics with access to references yield comparable performance.

| Ablation Setting | Overall Score |
|---|---|
| Essential-Only Rubrics | 34.9% |
| No Categorical Labels | 38.8% |
| No Pitfall Criteria | 37.2% |
| All Rubrics | 37.2% |

Table 3: Ablation results for elements of rubric design on HealthBench-1k (trained on HealthBench-3.5k subset with Qwen2.5-7B base policy). Rubrics generated using `o3-mini` with access to reference answers.

In-domain testing on HealthBench-1k amplifies RaR's gains: every instance-specific rubric-based method outperforms rubric-free baselines. Notably, even the weakest RaR variant significantly surpasses `Reference-Likert`, underscoring the advantage of structured supervision in subjective, open-ended domains like healthcare. We attribute this to the finer granularity and clarity rubrics provide in assigning rewards-especially when correctness is not binary and answers may vary in tone, completeness, or safety relevance.

Moreover, we find that rubric quality is crucial: synthetic rubrics generated with reference-answer guidance consistently outperform those generated without it. This highlights the importance of incorporating expert signal, whether via human-in-the-loop annotations or high-quality reference completions, for generating effective and aligned rubrics. Purely synthetic rubrics, while scalable, currently fall short in capturing the subtle criteria required for robust training in high-stakes domains.

**Elements of Rubric Design** This ablation study examines how the structure and weighting of synthetic rubrics affect downstream performance on HealthBench-1k. As shown in Table 3, rubrics that include a broader range of criteria outperform those limited to essential checks, suggesting that richer evaluation signals lead to better learning. Interestingly, we observe minimal performance differences when including rubric weights or pitfall criteria during training. One possible explanation is that synthetically generating effective pitfall criteria is inherently difficult, as it requires anticipating the most common or critical failure modes of the model, a task that often demands human intuition and domain expertise. As a result, these synthetic negative criteria may lack the specificity or relevance needed to meaningfully penalize undesirable responses.

**Impact of LLM Expertise on Rubric Quality** To assess how the capabilities of rubric-generating LLMs affect downstream performance, we generate synthetic rubrics without access to reference answers and use them to train policies on HealthBench. This isolates the effect of LLM quality on reference-free rubric utility. Specifically, we evaluate on the HealthBench-1k subset, using models trained on rubrics generated for the remaining 4k training examples from HealthBench.

As shown in Table 4, larger or more capable LLMs generally produce more effective rubrics, with GPT-4o yielding the best performance among reference-free models. However, all remain below the performance of rubrics generated with reference guidance (e.g., O3-mini with access to reference

| Rubric Generation Model | Overall Score |
|---|---|
| O3-mini (Rubrics with reference) | 35.9% |
| GPT-4o | 34.2% |
| GPT-4o-mini | 32.7% |
| O3-mini | 32.4% |
| Qwen-72B-Instruct | 32.7% |
| Qwen-32B-Instruct | 31.1% |
| Qwen-7B-Instruct | 31.9% |

Table 4: Policy performance on HealthBench-1k when trained with GRPO using rubrics generated by different LLMs *without* reference answers. GPT-4o-generated rubrics yield the strongest performance, though they still fall short of rubrics generated with expert (reference-guided) supervision. Smaller aligned models (e.g., GPT-4o-mini, O3-mini) remain competitive with larger open-weight models, underscoring the importance of alignment and reasoning ability in rubric generation.

| Baseline/Method | Overall Score |
|---|---|
| Qwen2.5-3B (Base Policy) | 4.13% |
| Direct-Likert | 13.74% |
| Reference-Likert | 17.95% |
| RaR-Implicit (ours) | **21.55%** |

Table 5: Performance on HealthBench-1k comparing methods trained on an alternative policy model (`Qwen2.5-3B`) shows that rubric-based reward training (`RaR-Implicit`) continues to outperform Likert-based baselines.

answers). Additionally, model attributes such as instruction tuning and reasoning capabilities play a key role in the effectiveness of rubric generation.

**Robustness to policy model choice** To assess whether our findings on rubrics as rewards remain agnostic to policy models, we repeat the training procedure using a smaller base model, `Qwen2.5-3B`. We train this model on the HealthBench-3k prompts using `Direct-Likert`, `Reference-Likert`, and `RaR-Implicit`, and evaluate on the held-out HealthBench-1k set. As shown in Table 5, the relative performance trends are consistent with the 7B model: `RaR-Implicit` achieves the strongest performance among the compared methods. This provides an additional data point suggesting that rubric-based rewards remain effective when applied to a smaller policy models.

## 7 RELATED WORK

**RLVR across domains** Reinforcement learning with verifiable rewards (RLVR) is expanding beyond math and code. GENERAL-REASONER trains on a 200k mixed corpus spanning physics, finance, and policy, and reports a ten-point gain on MMLU-Pro after GRPO fine-tuning (Ma et al., 2025). A follow-up extends RLVR to medicine, chemistry, psychology, and economics, showing that a single cross-domain reward model can supervise all four without task-specific tweaks (Su et al., 2025a). In healthcare, MED-RLVR applies similar methods to multiple-choice clinical QA, improving accuracy over supervised baselines while eliciting chain-of-thought from a 3B base model (Zhang et al., 2025). These results indicate steady progress, yet sparse signals, verifier reliability, and limited benchmark coverage remain open challenges.

**Rubrics for evaluation and training** Task-specific rubrics are increasingly used to evaluate LLMs in difficult-to-verify domains (Arora et al., 2025; Ruan et al., 2025; Hashemi et al., 2024; Pathak et al., 2025). Pathak et al. show that rubric-prompted LLM graders are more accurate and consistent than a question-agnostic checklist (Pathak et al., 2025). HEALTHBENCH scales this idea in medicine, pairing 48k clinician-written criteria with GPT-4 judges to score various axes (Arora et al., 2025). Beyond evaluation, rubrics are used to condition preference pairs for DPO (CPT; (Gallego, 2025)) and to guide checklist-based preference tuning in safety, instruction-following, and creative-writing settings (Viswanathan et al., 2025; Dineen et al., 2025; Kim et al., 2025). These lines of

work primarily use rubrics to grade outputs or to condition preference data, often in non-reasoning domains such as safety, instruction following, or creative writing. In contrast, we use rubric criteria directly as reward signals for on-policy RL in expert-reasoning and real-world domains.

**Learning from feedback signals**  RLHF trains policies with large numbers of human comparisons, which introduces subjectivity and can lead to reward hacking (Ouyang et al., 2022). RLVR reduces these issues by using programmatic checks, from exact matches on GSM8K and MATH to mixed-domain verifiers in GENERAL-REASONER and CROSS-DOMAIN RLVR (Ma et al., 2025; Su et al., 2025a), although signals can be sparse. Process supervision (Lightman et al., 2023) provides denser guidance via step-level labels, and MCTS-generated annotations or generative reward models such as THINKPRM improve performance, but with high annotation cost (Li et al., 2025; Khalifa et al., 2025). Rubric-based RL finds a middle ground by turning multiple rubric criteria into structured verifiers and using their scalar scores as denser rewards.

## 8   CONCLUSION

We introduced **Rubrics as Rewards (RaR)**, a framework for post-training language models using structured, checklist-style rubrics as reward signals. By decomposing response evaluation into transparent, multi-criteria objectives—both subjective and objective—RaR provides a modular and interpretable alternative to preference-based methods. Our experiments demonstrate that rubric-guided training achieves strong performance across domains, significantly outperforming Likert-based baselines and matching or exceeding the performance of reference-based reward generation. Additionally, we show that RaR improves alignment with human preferences while reducing dependence on large judge models.

## 9   LIMITATIONS AND FUTURE WORK

Our work focuses on medicine and science to enable controlled experiments. This choice allows us to run controlled experiments, but broader validation across dialogue, tool use, or other agentic tasks remains an important direction. We evaluate only two reward aggregation strategies, implicit and explicit, since they capture complementary extremes of flexibility and control; future work could explore more advanced ways of combining rubric criteria, such as learning continuous weights for each criterion or dynamically adjusting weights over the course of training to mimic a curriculum (e.g., prioritizing essential correctness early, then gradually emphasizing more subtle qualities like style or safety). Finally, we use off-the-shelf LLMs as judges for accessibility and reproducibility; exploring specialized evaluators with stronger reasoning or generative reward models may yield further gains.

### REPRODUCIBILITY STATEMENT

Our datasets *RaR-Medicine* and *RaR-Science* datasets with per-prompt rubrics are released at `https://huggingface.co/collections/ScaleAI/rar`. We document data construction and rubric synthesis in Sections 2 and 3, with full prompt templates in Appendices A.6, A.8, and A.10. Training settings for on-policy GRPO (model, batch size, schedule, hardware) appear in Section 4.2 and Table 12. All judge prompts used to compute rewards for rubric-guided and Likert baselines are in Appendix A.7. Evaluation protocols, metrics, and prompts are detailed in Section 4.5 and Appendix A.5; we report means with 95% confidence intervals and use the open-source `simple-evals` framework for evaluation.

### ACKNOWLEDGMENTS

We thank Qin Lyu, Nikhil Barhate, and Zijian Hu for supporting this research through the development of the in-house post-training platform RLXF. We also thank Robert Vacareanu for valuable early feedback and discussions.

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

## A  APPENDIX

### A.1  USE OF LLMS

We use LLMs for minor editorial support, including grammar, spelling, punctuation, latex formatting and word choice corrections, with usage limited to enhancing readability and clarity.

### A.2  DETAILS OF RAR-MEDICINE DATASET

The following is illustrates an example from the RaR-Medicine dataset with more details about statistics and categorical distribution of rubrics listed in Table 6 and 8.

---

**RaR-Medicine Example**

**Question.** A 50-year-old male patient weighs 65 kg with a pH of 7.05, PCO2 of 15 mmHg, HCO3 of 5 mEq/L, and a base deficit of -40 mEq/L. How much sodium bicarbonate should be administered in the first 4 hours to correct his metabolic acidosis?

**Reference Answer.** To address the severe metabolic acidosis in this 50-year-old patient (weight 65 kg) with a pH of 7.05, it is crucial to approach treatment with caution to avoid complications from overcorrection. The base deficit is -40 mEq/L, which indicates a significant acid-base disturbance.
Typically, the formula for bicarbonate administration is:

$$\text{Bicarbonate required} = \text{Base Deficit(mEq/L)} \times \text{Body Weight(kg)} \times 0.3$$

Applying the values:

$$40\,\text{mEq/L} \times 65\,\text{kg} \times 0.3 = 780\,\text{mEq}$$

However, it is standard practice to administer a partial correction initially, aiming for safety and gradual improvement. Therefore, administering about 150 mEq of sodium bicarbonate over the first 4 hours is a prudent approach. This amount aligns with safe clinical guidelines for initial treatment in severe metabolic acidosis, ensuring the patient starts on the path to stabilization without risking harm from rapid correction.

**Rubric (checklist and weights).**

- **Bicarbonate Calculation** ($w$=5): Essential Criteria: The response must correctly identify and apply the formula (Base Deficit × Body Weight × 0.3) to determine the bicarbonate requirement.

- **Safe Dosing Recommendation** ($w$=5): Essential Criteria: The response must state a clear recommendation of administering about 150 mEq of sodium bicarbonate over the first 4 hours.

- **Partial Correction Justification** ($w$=4): Important Criteria: The response should explain that only a partial correction is administered initially to avoid complications from rapid overcorrection.

- **Step-by-Step Calculation** ($w$=3): Important Criteria: The response must detail the calculation steps, showing that 40 mEq/L × 65 kg × 0.3 equals 780 mEq, before noting the adjusted dose for safety.

- **Base Deficit Interpretation** ($w$=2): Optional Criteria: The response may mention that the base deficit of -40 mEq/L indicates severe metabolic acidosis requiring cautious treatment.

- **Patient Data Accuracy** ($w$=3): Important Criteria: The response must accurately incorporate the patient's weight of 65 kg and his critical pH, PCO2, and HCO3 values into the explanation.

- **Avoid Overcorrection Risk** ($w$=-1): Pitfall Criteria: Does not mention the risks associated with rapid overcorrection of metabolic acidosis if only the full calculated bicarbonate amount is administered.

---

| Metric | Value |
|---|---|
| Total examples | 20,166 |
| Avg. rubrics per question | 7.5 |
| Avg. question length (words) | 45.0 |

Table 6: Aggregate statistics for the RaR-Medicine dataset (train and validation) dataset.

| Rubric Type | Count | Percent |
|---|---|---|
| Important | 52,748 | 34.1 |
| Essential | 47,584 | 30.7 |
| Optional | 34,261 | 22.1 |
| Pitfall | 20,215 | 13.1 |

Table 7: Rubric-type distribution across all 20,166 examples.

| Topics | Count | Percent |
|---|---|---|
| **Total examples** | **20,166** | **100.0** |
| Medical Diagnosis | 10,147 | 50.3 |
| Medical Treatment | 3,235 | 16.0 |
| Medical Knowledge | 2,557 | 12.7 |
| Medical Diag. and Mngmnt | 2,033 | 10.1 |
| Medical Biology | 770 | 3.8 |
| Other | 428 | 2.1 |
| Medical Ethics | 377 | 1.9 |
| Health Physics | 276 | 1.4 |
| Epidemiology & Pub. Health | 216 | 1.1 |
| General Medicine | 113 | 0.6 |
| Forensic Medicine | 14 | 0.1 |

Table 8: Distribution of topics in the medical training and validation dataset

## A.3 DETAILS OF RAR-SCIENCE DATASET

This section shows an illustrative example from the RaR-Science dataset with more details about statistics and categorical distibution of rubrics listed in Table 9, 10, 11.

---

**RaR-Science Example**

**Question.** Determine the solubility of boric acid ($H_3BO_3$) in ethanol ($C_2H_5OH$) compared to its solubility in benzene ($C_6H_6$), considering the principles of 'likes dissolve likes' and the role of $K_{sp}$ values. Explain your reasoning and provide examples of how immulcifiers could affect the solubility of substances in different solvents.

**Reference Answer.** Boric acid is more soluble in ethanol than in benzene.

**Rubric (checklist and weights).**

- **Correct Solubility Direction** ($w$=5): Essential Criteria: The response must clearly identify that boric acid is more soluble in ethanol than in benzene.

- **Polarity Principle** ($w$=5): Essential Criteria: The answer should explain how the 'like dissolves like' principle applies by contrasting the polar nature of ethanol with the non-polar character of benzene.

- **Ksp Context** ($w$=4): Important Criteria: The response should account for the role of Ksp values, discussing their typical relevance to solubility even though boric acid is a covalent compound rather than an ionic one.

- **Immulcifier Explanation** ($w$=4): Important Criteria: The answer should explain how immulcifiers could modify solubility, providing an example of their effect on solvation in different solvents.

- **Chemical Properties** ($w$=4): Important Criteria: The response should analyze the inherent chemical properties of boric acid and the solvents to justify the observed solubility differences.

- **Avoid Ionic Assumptions** ($w$=-1): Pitfall Criteria: The answer must not incorrectly assume that Ksp values for ionic compounds directly determine the solubility of a covalent acid such as boric acid.

- **Enhanced Detail** ($w$=2): Optional Criteria: The response may include additional examples or a concise explanation of solvation dynamics to further illustrate how solubility is influenced.

| Metric | Value |
|---|---|
| Total examples | 20,625 |
| Avg. rubrics per question | 7.5 |
| Avg. question length (words) | 52.6 |

Table 9: Aggregate statistics for the full medical (train and validation) dataset.

| Rubric Type | Count | Percent |
|---|---|---|
| Important | 52,315 | 34.8 |
| Essential | 42,739 | 28.4 |
| Optional | 33,622 | 22.3 |
| Pitfall | 21,808 | 14.5 |

Table 10: Rubric-type distribution across all 20 625 examples.

## A.4 TRAINING DETAILS

The training hyperparameters are described in Table 12.

| Topics | Count | Percent |
|---|---|---|
| **Total examples** | **20625** | **100.0** |
| General Chemistry | 3163 | 15.3 |
| Quantum Mechanics | 3158 | 15.3 |
| Physical Chemistry | 2761 | 13.4 |
| Statistical Mechanics | 2530 | 12.3 |
| Organic Chemistry | 2059 | 10.0 |
| General Physics | 1439 | 7.0 |
| Condensed Matter Physics | 1387 | 6.7 |
| Genetics | 1378 | 6.7 |
| Molecular Biology | 815 | 4.0 |
| Astrophysics | 409 | 2.0 |
| Inorganic Chemistry | 407 | 2.0 |
| Analytical Chemistry | 398 | 1.9 |
| Electromagnetism | 239 | 1.2 |
| Optics | 143 | 0.7 |
| High Energy Physics | 116 | 0.6 |
| Electromagnetic Theory | 105 | 0.5 |
| Electromagnetics | 72 | 0.3 |
| Relativistic Mechanics | 46 | 0.2 |

Table 11: Distribution of topics in the STEM training and validation dataset

Rubric-based evaluation consistently yields stronger policies across all judge sizes. The most pronounced improvement appears with Qwen-7B-Instruct (+0.047), where rubric guidance lifts it from weakest to nearly matching larger models. Additionally, rubric-based scores are more tightly clustered (0.250–0.279) than those from Likert-only judges (0.220–0.254), indicating improved consistency.

These results suggest that rubrics help smaller judges approximate high-quality supervision by breaking evaluation into interpretable, binary criteria. This structured approach mitigates scale-related limitations, enabling more reliable reward modeling even with limited-capacity evaluators.

| **Hyperparameters** | |
|---|---|
| num_rollouts_per_prompt | 16 |
| batch_size (effective) | 96 |
| sampling_temperature | 1.0 |
| warmup_ratio | 0.1 |
| learning_rate | 5.0e-06 |
| lr_scheduler_type | constant_with_warmup |
| max_length | 3584 |
| num_train_steps | 300 |

Table 12: GRPO hyperparameter settings for Medical and Science domains.

| Judge Model | RaR-Implicit | Direct-Likert |
|---|---|---|
| GPT-4o-mini | 27.9% | 25.3% |
| Qwen-32B-Instruct | 26.2% | 25.4% |
| Qwen-14B-Instruct | 25.0% | 24.9% |
| Qwen-7B-Instruct | 26.7% | 22.0% |

Table 13: Judge quality comparison: rubric-based evaluation vs pure Likert scoring on synthetic medical rubrics.

## A.5 EVALUATION PROMPTS

---

**GPQA Evaluation Prompt**

Determine whether the following model response matches the ground truth answer.

```
## Ground truth answer##: Option {correct_answer} or {correct_answer_text}
## Model Response ##: {response_text}
```

A response is considered correct if it's final answer is the correct option letter (A, B, C, or D), or has the correct answer text.
Please respond with only "Yes" or "No" (without quotes). Do not include a rationale.

---

## A.6 PREDEFINED STATIC RUBRICS

---

**Predefined Static Rubrics for RaR-Static Method**

- The response contains correct information without factual errors, inaccuracies, or hallucinations that could mislead the user.
- The response fully answers all essential parts of the question and provides sufficient detail where needed.
- The response is concise and to the point, avoiding unnecessary verbosity or repetition.
- The response effectively meets the user's practical needs, provides actionable information, and is genuinely helpful for their situation.

---

## A.7 LLM-JUDGE PROMPTS

---

**Prompt for RAR-IMPLICIT Method**

**System Prompt:**
You are an expert evaluator. Given a user prompt, a generated response, and a list of quality rubrics, please rate the overall quality of the response on a scale of 1 to 10 based on how well it satisfies the rubrics.
Consider all rubrics holistically when determining your score. A response that violates multiple rubrics should receive a lower score, while a response that satisfies all rubrics should receive a higher score.
Start your response with a valid JSON object that starts with "```json" and ends with "```". The JSON object should contain a single key "rating" and the value should be an integer between 1 and 10.
Example response:
```json
{
"rating": 7
}```

**User Prompt Template:**
Given the following prompt, response, and rubrics, please rate the overall quality of the response on a scale of 1 to 10 based on how well it satisfies the rubrics.

```
<prompt>
{prompt}
</prompt>

<response>
{response}
</response>

<rubrics>
{rubric_list_string}
</rubrics>
```
Your JSON Evaluation:

---

**Prompt for DIRECT-LIKERT Baseline**

**System Prompt:**
You are an expert evaluator. Given a user prompt and a generated response, please rate the overall quality of the response on a scale of 1 to 10, where 1 is very poor and 10 is excellent.
Start your response with a valid JSON object that starts with "```json" and ends with "```". The JSON object should contain a single key "rating" and the value should be an integer between 1 and 10.
Example response:
```json
{
"rating": 8
}```

**User Prompt Template:**
Given the following prompt, and response, please rate the overall quality of the response on a scale of 1 to 10.

```
<prompt>
{prompt}
</prompt>

<response>
{response}
</response>
```
Your JSON Evaluation:

---

**Prompt for REFERENCE-LIKERT Baseline**

**System Prompt:**
You are an expert evaluator. Given a user prompt, a reference response, and a generated response, please rate the overall quality of the generated response on a scale of 1 to 10 based on how well it compares to the reference response.
Consider factors such as accuracy, completeness, coherence, and helpfulness when comparing to the reference. The reference response represents a high-quality answer that you should use as a benchmark.
Start your response with a valid JSON object that starts with "```json" and ends with "```". The JSON object should contain a single key "rating" and the value should be an integer between 1 and 10.
Example response: ```json
{
"rating": 8
}```

**User Prompt Template:** Given the following prompt, reference response, and generated response, please rate the overall quality of the generated response on a scale of 1 to 10 based on how well it compares to the reference.

```
<prompt>
{prompt}
</prompt>

<reference_response>
{reference}
</reference_response>

<generated_response>
{response}
</generated_response>
```

Your JSON Evaluation:

---

## A.8  SYNTHETIC PREFERENCE SET GENERATION

We leverage the publicly-released HEALTHBENCH (Arora et al., 2025) corpus, which contains **5,000** health-related prompts accompanied by expert-written answers. Of these, **4,203** datapoints already include an *ideal* completion vetted by licensed clinicians. For every such prompt–ideal pair we automatically generate a *perturbed* counterpart using o3 with the structured template shown below. The template forces the model to (i) spell out a [reasoning] plan for degrading quality, (ii) emit the degraded [perturbed_completion], and (iii) log exact [chunks_added] and [chunks_removed]. Perturbations are accepted only after manual screening confirms that they are *objectively worse*, along at least one axis of medical accuracy, completeness, clarity, safety, specificity, structure, or tone, while remaining coherent and free of dangerous advice. We further exclude the prompts from HealthBench-1k used for ablations. This procedure produces a balanced evaluation set of **3,027 preferred** and **3,027 perturbed** responses (6,054 total), which we use in the rubric-versus-Likert experiments in Section 5. The prompt used for this generation is detailed in Figure A.10.

## A.9  JUDGE QUALITY IMPACTS ON POST-TRAINING

We assess whether rubric-guided evaluation improves judge effectiveness compared to rubric-free Likert scoring when used for GRPO training. Table 13 reports judge accuracy on synthetic medical data, with all policies trained using Qwen2.5-7B and varying judge models.

---

**Prompt for Synthetic Rubrics Generation: Medical Domain**

You are an expert rubric writer. Your job is to generate a self-contained set of evaluation criteria ("rubrics") for judging how good a response is to a given question. Rubrics can cover aspects of a response such as, but not limited to, factual correctness, ideal-response characteristics, style, completeness, helpfulness, harmlessness, patient-centeredness, depth of reasoning, contextual relevance, and empathy. Each item must be self-contained – non expert readers should not need to infer anything or consult external information. Begin each description with its category: "Essential Criteria: ...", "Important Criteria: ...", "Optional Criteria: ...", or "Pitfall Criteria: Does not mention ...".

Inputs:

- `question`: The full question text.
- `reference_answer`: The ideal answer, including any specific facts, explanations, or advice.

Total items:

- Choose 7–20 rubric items based on the complexity of the question.

Each rubric item:

- **title** (2–4 words).
- **description**: One sentence starting with its category prefix that explicitly states exactly what to look for. For example:
  - Essential Criteria: Identifies non-contrast helical CT scan as the most sensitive modality for ureteric stones.
  - Pitfall Criteria: Does not mention identifying (B) as the correct answer.
  - Important Criteria: Explains that non-contrast helical CT detects stones of varying sizes and compositions.
  - Optional Criteria: States "The final answer is (B)" or similar answer choice formatting.
- **weight**: For Essential/Important/Optional, use 1–5 (5 = most important); for Pitfall, use –1 or –2.

Category guidance:

- **Essential**: Critical facts or safety checks; if missing, the response is invalid (weight 5).
- **Important**: Key reasoning, completeness, or clarity; strongly affects quality (weight 3–4).
- **Optional**: Helpful style or extra depth; nice to have but not deal-breaking (weight 1–2).
- **Pitfall**: Common mistakes or omissions specific to this prompt—identify things a respondent often forgets or misstates. Each Pitfall description must begin with "Pitfall Criteria: Does not mention ..." or "Pitfall Criteria: Recommends ..." and use weight –1 or –2.

To ensure self-contained guidance:

- When referring to answer choices, explicitly say "Identifies (A)", "Identifies (B)", etc., rather than vague phrasing.
- If the format requires a conclusion like "The final answer is (B)", include a rubric item such as:
  - Essential Criteria: Includes a clear statement "The final answer is (B)".
- If reasoning should precede the answer, include a rubric like:
  - Important Criteria: Presents the explanation before stating the final answer.
- If brevity is valued, include a rubric like:
  - Optional Criteria: Remains concise and avoids unnecessary detail.
- If the question context demands mention of specific findings, include that explicitly (e.g., "Essential Criteria: Mentions that CT does not require contrast").

Output: Provide a JSON array of rubric objects. Each object must contain exactly three keys—title, description, and weight. Do not copy large blocks of the question or reference_answer into the text. Each description must begin with its category prefix, and no extra keys are allowed.

Now, given the question and reference_answer, generate the rubric as described. The reference answer is an ideal response but not necessarily exhaustive; use it only as guidance.

---

**Prompt for Synthetic Rubric Generation: Science Domain**

You are an expert rubric writer for science questions in the domains of Biology, Physics, and Chemistry. Your job is to generate a self-contained set of evaluation criteria ("rubrics") for judging how good a response is to a given question in one of these domains. Rubrics can cover aspects such as factual correctness, depth of reasoning, clarity, completeness, style, helpfulness, and common pitfalls. Each rubric item must be fully self-contained so that non-expert readers need not consult any external information.

**Inputs:**

- `question`: The full question text.
- `reference_answer`: The ideal answer, including any key facts or explanations.

**Total items:**

- Choose 7–20 rubric items based on question complexity.

Each rubric item must include exactly three keys:

1. **title** (2–4 words)
2. **description**: One sentence beginning with its category prefix, explicitly stating what to look for. For example:
   - Essential Criteria: States that in the described closed system, the total mechanical energy (kinetic plus potential) before the event equals the total mechanical energy after the event.
   - Important Criteria: Breaks down numerical energy values for each stage, demonstrating that initial kinetic energy plus initial potential energy equals final kinetic energy plus final potential energy.
   - Optional Criteria: Provides a concrete example, such as a pendulum converting between kinetic and potential energy, to illustrate how energy shifts within the system.
   - Pitfall Criteria: Does not mention that frictional or air-resistance losses are assumed negligible when applying conservation of mechanical energy.
3. **weight**: For Essential/Important/Optional, use 1–5 (5 = most important); for Pitfall, use –1 or –2.

**Category guidance:**

- Essential: Critical facts or safety checks; omission invalidates the response.
- Important: Key reasoning or completeness; strongly affects quality.
- Optional: Nice-to-have style or extra depth.
- Pitfall: Common mistakes or omissions; highlight things often missed.

**Format notes:**

- When referring to answer choices, explicitly say "Identifies (A)", "Identifies (B)", etc.
- If a clear conclusion is required (e.g. "The final answer is (B)"), include an Essential Criteria for it.
- If reasoning should precede the final answer, include an Important Criteria to that effect.
- If brevity is valued, include an Optional Criteria about conciseness.

**Output:** Provide a JSON array of rubric objects. Each object must contain exactly three keys—title, description, and weight. Do not copy large blocks of the question or reference_answer into the text. Each description must begin with its category prefix, and no extra keys are allowed.

Now, given the question and reference_answer, generate the rubric as described. The reference answer is an ideal response but not necessarily exhaustive; use it only as guidance.

## A.10 Perturbed Dataset Generation

---

**Prompt for Health Response Perturbation**

You are tasked with creating perturbed versions of health-related responses to generate training data for evaluating model robustness. Given an original health prompt and an ideal completion, you must generate a perturbed version that is strictly worse than the ideal response but still plausible.

**Input Format:**

`[prompt]`: prompt

`[ideal_completion]`: ideal_completion

**Task Instructions:** For the perturbed version, follow this exact process:

`[reasoning]`: Explain your strategy for making the response worse. Consider these perturbation approaches:

- Medical Accuracy: Introduce subtle medical inaccuracies or outdated information - Completeness: Remove crucial information or context - Clarity: Make explanations less clear or more confusing - Safety: Reduce appropriate medical disclaimers or caveats - Specificity: Replace specific advice with vague generalizations - Structure: Disorganize the logical flow of information - Tone: Make the response less empathetic or more dismissive

`[perturbed_completion]`: Create the modified response based on your reasoning strategy.

`[chunks_added]`: List any new text segments you added (even if they're incorrect or unhelpful)

`[chunks_removed]`: List any text segments you removed from the ideal response

**Quality Guidelines:**

- Ensure Degradation: Each perturbed response must be objectively worse than the ideal response - Maintain Plausibility: The perturbed responses should still seem like reasonable attempts (not obviously nonsensical) - Vary Approaches: Use different perturbation strategies for each version - Track Changes Precisely: Be specific about what text was added or removed - Health Context Sensitivity: Be mindful that health misinformation can be harmful - focus on subtle degradations rather than dangerous advice

**Example Perturbation Strategies:**

- Remove important safety warnings or medical disclaimers - Add outdated medical information that was once accepted but is now known to be incorrect - Remove specific dosage information or timing details - Add overly general statements that replace specific guidance - Remove context about when to seek professional medical help - Add confusing or contradictory information - Remove step-by-step instructions and replace with vague advice - Add unnecessarily complex medical jargon without explanation

---

## A.11 Additional Benchmark Analysis

We provide additional results for Science domain by evaluating baselines and methods trained on RaR-Science dataset by evaluating on the Chemistry, Physics and Biology subset of MMLU-Pro Wang et al. (2024b). The results are shown in Table A.11.

| Task | Qwen2.5-8B Base | Simple-Likert | Reference-Likert | Explicit-RaR | Implicit-RaR |
|---|---|---|---|---|---|
| Chemistry | 45.49% | 51.15% | 50.88% | 54.06% | 52.47% |
| Physics | 49.50% | 55.04% | 51.81% | 55.58% | 54.27% |
| Biology | 64.30% | 68.76% | 66.39% | 68.34% | 68.90% |
| **Micro Avg.** | 51.43% | 56.77% | 54.80% | 57.94% | 56.96% |
| **Macro Avg.** | 53.10% | 58.32% | 56.36% | 59.33% | 58.55% |

Table 14: **MMLU-Pro (Science subset) Regression Analysis.** Performance of different reward strategies on the MMLU-Pro benchmark's Science subset.

