# OpenReview forum: "Rubrics as Rewards: Reinforcement Learning Beyond Verifiable Domains"
_ICLR.cc/2026/Conference — ICLR 2026 Poster_

### Official Review · Reviewer_dV3e · 2025-10-25

**Soundness:** 2
**Presentation:** 4
**Contribution:** 2
**Rating:** 2
**Confidence:** 3

**Summary:**

This paper presents Rubrics as Rewards (RaR) that extends reinforcement learning with verifiable rewards.  It uses LLMs to generate checklist-style rubrics for each instance from the reference answer, and uses an aggregated score from these rubrics as the final rewards.  It also proposes 2 aggregation methods: explicit (with human assigned weights) and implicit (with LLM-as-a-judge to generate a score). Experiments show that their methods can out-perform the baseline methods such as likert-based rewards on benchmarks like HealthBench and GPQA Diamond.

**Strengths:**

RaR shows consitent performance gain over the baseline methods without needing further human annotation.  It well-aligns the design of HealthBench, which also evaluates the responses with its own rubrics.

This paper does some ablation studies and analysis, such evaluating RaR using human preference metric. On this metric, we can see that RaR can generate more human-preferred responses.

**Weaknesses:**

1. The performance improvement is marginal: The paper claims a 31% relative performance gain, which is partially true. On fig 2, not all baselines are fair comparison, and they might not be trained on HealthBench or not have access to the reference answers. Comparing RaR to reference-likert baseline, its performance gain is marginal on HealthBench (2.3 pts) and not significant on GPQA-Diamond (1.1 pts and error bars overlap).
2. Limited experiments: This paper only experiment on HealthBench and GPQA-Diamond. The former dataset is designed specifically with rubrics, thus being a good candidate dataset. But there are many multi-choice datasets similar to GPQA, and the authors should report results on these datasets to reduce the dataset noise.
3. The main point of this paper is that rubics-based reward is better than direct prompting. However, explicit aggragation of rubric scores cannot out-perform implicit aggregation, while the "implicit aggregation" can be considered as a more delicate prompt that inputs question, reference answer, and response to LLM to get an overall. It has no difference to the Likert-based LLM-as-a-judge method in nature.
4. More baselines can be introduced, such as fine-tuning with reinforcement learning and question-agnostic rubrics.

**Questions:**

1. If you do not use LLM to generate rubrics but directly use the rubrics of HealthBench, would that further improve the performance of RaR on HealthBench?

---

> ### Author Response · Authors · 2025-11-21
> **Response to Reviewer dV3e**
>
> We thank the reviewer for their comments and address the key points below, focusing on clarifying the factual premises.
>
> > On fig 2, not all baselines are fair comparison, and they might not be trained on HealthBench or not have access to the reference answers.
>
> To clarify, none of the methods or baselines in the main results (Figure 2) are trained on HealthBench dataset rubrics/reference answers. They are trained on RaR-Medical dataset as described in Section 4.1.
>
>
> > performance improvement is marginal between Reference-Likert baseline and RaR
>
> We clarify why the gains are meaningful on each benchmark with some further results.
> HealthBench. The +2.3 absolute gain over Reference Likert is meaningful in HealthBench’s rubric-based evaluation setting. Scores are normalized to 0–1 over physician-authored, multi-criteria rubrics for multi-turn clinical chats. A couple of points corresponds to satisfying several additional clinically important rubric items per conversation, so this reflects a real quality lift in the exact behavior RaR is trained to optimize.
>
> We also see a larger gap in a more realistic ablation where we train directly on HealthBench conversations (3k train, 1k eval): Reference Likert is 31.7% vs RaR Implicit 35.9%, a **+4.2 gain**. This regime is harder for reference-matching rewards because real conversations often have unstructured answers, making “match the reference” noisy. Rubrics stay explicit about what matters, which is where RaR helps most. This larger lift is especially relevant for the real-world RLVR use cases our paper targets.
> GPQA-Diamond. The main table confidence intervals came from our initial 10-run evaluation. GPQA-Diamond has only 198 questions, so even 10 runs leaves visible variance and can make intervals appear to overlap. We therefore reran GPQA with 100 independent runs to  to tighten the confidence intervals: Direct Likert 0.3399 ± 0.0054, Reference Likert 0.3426 ± 0.0051, Implicit RaR 0.3696 ± 0.0024. With a larger sample set, Implicit RaR is **+2.7 points** over Reference Likert and the intervals no longer overlap. The earlier “not significant” impression was an artifact of low effective sample size, not an absence of improvement.
>
> | Method/Baseline      | Score (mean ± 95% CI over 100 runs) |
> |-----------------------|------------------------|
> | Direct-Likert         | 0.3399 ± 0.0054        |
> | Reference-Likert      |    0.3426 ± 0.0051    |
> | RaR-Explicit          | 0.3570 ± 0.0045        |
> | RaR-Implicit          | 0.3696 ± 0.0024        |
>
>
> > there are many multi-choice datasets similar to GPQA, and the authors should report results on these datasets to reduce the dataset noise.
>
> Although many multiple-choice datasets exist for RLVR evaluation, few match GPQA-Diamond’s difficulty (PhD-written, Google-proof, oversight-motivated), and even fewer are available within the science domain, which is the setting of our experiments. Most MCQ benchmarks (e.g., MMLU/-Pro, SAT Science, ARC, AMC) focus on broad factual recall or single-domain skills and are substantially easier.
>
> Also, it is worth restating why we include an RLVR-style MCQ benchmark at all. GPQA-Diamond is not meant to be our main score target. We use it to check the robustness of the evaluation format. Specifically, we want to show that rubric-trained rewards learned from open-ended, rubric-scored data still transfer to a different verifiable setting like MCQs, and do not collapse when the evaluation changes. The true gains of RaR are best measured on rubric-based evaluations like HealthBench, which test multiple dimensions of quality. The MCQ results are a complementary sanity check, not the primary objective.
>
> That said, to address the breadth concern, we add a regression analysis on the Biology, Chemistry, and Physics subset of MMLU-Pro in Appendix A11. RaR shows no regressions and slightly edges out the Direct-Likert baseline on Micro Avg (56.96% for RaR-Implicit and 57.94% for RaR-Explicit vs 56.77% for Direct-Likert). Reference-Likert is notably lower (54.80%), reinforcing that unstructured reference supervision is more prone to regression. Overall, these results suggest RaR’s gains are not benchmark-specific.

---

> ### Author Response · Authors · 2025-11-21
> **Response to Reviewer dV3e**
>
> > implicit aggregation has no difference to the Likert-based LLM-as-a-judge method in nature
>
> We appreciate the comment, but it frames Implicit RaR as “nicer Likert prompting,” overlooking that the judge is conditioned on an instance-specific rubric and thus provides a different form of reward supervision.
> Empirically, we also find that the gains are driven by rubric structure, not prompt polish. To illustrate this, we recall that Figure 3 studies judge alignment across model scales and finds that rubric-conditioned judging (RaR, orange) consistently improves alignment with human preferences relative to pure Likert scoring (blue). Even synthetic, question-agnostic rubrics (green) beat Direct Likert, while expert-grounded rubrics perform best, and the advantage is largest for smaller judges. This indicates the rubric checklist, even when used implicitly, provides a more reliable evaluation signal than holistic Likert ratings.
>
> > more baselines can be introduced (question-agnostic rubrics, fine-tuning with RL)
>
>  The suggested “question-agnostic rubric” baseline is already in the paper. Section 4.4 defines Predefined-RaR, which uses a fixed set of rubrics for all prompts (detailed in Appendix A6).
> More broadly, we believe the current suite of baselines already covers the relevant ground for our claim. All methods, including ours, are fine tuned with GRPO, which is itself an RL algorithm. Within this shared RL setup, we compare strong alternative reward constructions (Direct Likert, Reference Likert, RaR Explicit, RaR Implicit), which cleanly isolates the effect of rubric conditioned rewards. Adding further “fine tuning with RL” baselines that switch to a different RL algorithm or reward model would be largely orthogonal to the paper’s focus.
>
> > Q1: If you do not use LLM to generate rubrics but directly use the rubrics of HealthBench, would that further improve the performance of RaR on HealthBench?
>
> Yes, directly using HealthBench rubrics improves performance of RaR. This analysis is already described in our first ablation (Table 2, Lines 369–377). To recap, we partition HealthBench into train and test subsets and evaluate RaR using the human-authored HealthBench rubrics as supervision. We observe that both structured rubrics (human-authored or LLM-generated) outperform other reward strategies by a substantial margin, with human-authored rubrics yielding more than 3 percentage-point improvement over the Reference Likert baseline. This ablation shows that rubric quality matters, and rubrics that capture human expertise outperform generic synthetic rubrics.

---

> > ### Comment · Reviewer_dV3e · 2025-11-24
> > **Reviewer's comments**
> >
> > Thanks for your clarification.  I acknowledge that I made some mistakes on my initial reviews about the interpretation of results, and I am willing to raise my rating to 4.

---

> > > ### Author Response · Authors · 2025-11-26
> > > **Acknowledgment to Reviewer dV3e**
> > >
> > > Thank you for taking the time to review our rebuttal. If you have any further questions or suggestions, please feel free to contact us at any time.

---

### Official Review · Reviewer_19Mf · 2025-10-30

**Soundness:** 2
**Presentation:** 2
**Contribution:** 2
**Rating:** 6
**Confidence:** 2

**Summary:**

This paper introduces Rubrics as Rewards (RaR), a framework that extends Reinforcement Learning with Verifiable Rewards (RLVR) to domains where correctness is multifaceted and not easily verifiable by a single binary signal. The authors create two datasets, RaR-Medicine and RaR-Science, by generating rubrics from reference answers using strong LLMs. Evaluations on HealthBench (rubric-based) and GPQA-Diamond (multiple-choice) show that RaR, particularly the Implicit variant, outperforms strong baselines like Direct-Likert and Reference-Likert scoring.

**Strengths:**

The paper successfully bridges a critical gap between RLVR and preference-based RLHF by introducing a structured, multi-criteria intermediate: the rubric. The method demonstrates consistent and significant improvements over strong baselines across two challenging domains medicine and science. The relative gains of up to 31% on HealthBench are substantial.

**Weaknesses:**

The entire RaR pipeline's success is contingent on the quality of the synthetically generated rubrics. While ablations show the importance of using reference answers, the inherent limitations of LLM-based generation are not fully addressed. The performance ceiling is thus tied to the rubric generator's capabilities.
The method requires generating a unique, detailed rubric for each of the 20k+ training instances and then using an LLM judge to evaluate 16 rollouts per prompt. This is computationally expensive and may be prohibitive for larger-scale applications without significant optimizations.
The generalizability of RaR to more creative, open-ended, or highly subjective tasks (e.g., essay writing, dialogue, creative design) remains an open question.
The two aggregation strategies (Explicit and Implicit) are static. The paper does not explore more dynamic or learned weighting schemes that could potentially adapt during training.
While Implicit Aggregation performs best, it also makes the reward signal less interpretable. Did you observe any cases where the implicit judge failed to properly incorporate the rubric, and could this opaqueness be a risk in safety-critical domains like medicine?

**Questions:**

See weakness above.

---

> ### Author Response · Authors · 2025-11-21
> **Response to Reviewer 19Mf**
>
> We thank the reviewer for highlighting the strengths of our work; specifically the introduction of rubric-based reward signals as a method to **bridge critical gap** between RLVR and preference-based RLHF, the strong empirical gains on HealthBench and GPQA-Diamond, and the **consistent improvements over competitive baselines**.
>
> Below, we address the concerns raised.
>
>
> > Success depends on the quality of synthetically generated rubrics; limitations of LLM-generated rubrics not fully addressed.
>
> *Clarification.* We agree that rubric quality is important. Our empirical results already highlight this: Table 2 shows that rubrics generated with expert-grounded reference answers outperform rubrics generated without them. This is precisely why, in this work, we deliberately use reference answers as high-quality supervision proxies.
>
>
> *Why synthetic rubrics are used.* Currently, no human-written training dataset of per-instance rubrics exists for reasoning tasks in medicine or science. Studying structured multi-criteria rewards at scale therefore requires synthetic generation. We explicitly state this limitation and discuss future extensions with human-authored rubrics (Sec. 3.2 and Sec. 9).
>
>
> *Future extensibility.* To enable precisely the type of optimization the reviewer envisions, we are releasing all rubrics publicly. This allows future work to train new verifiers, explore alternative rubric-generation strategies, or incorporate human-written rubrics when available.
>
>
> > The method requires generating a unique rubric + 16 judged rollouts per instance; computationally expensive.
>
> *Clarification.* All large-scale on-policy RL is computationally heavy, this is inherent to GRPO and RLVR pipelines broadly. RaR does not add additional model training or inference beyond what is already required in standard GRPO setups: only the rubric generation step is additional, and this is a one-time preprocessing cost.
>
>
> > Generalizability to open-ended subjective tasks (dialogue, creative writing) remains unclear.
>
> We want to highlight that the HealthBench benchmark is not structured reasoning. It contains multi-turn real-world patient–provider conversations which is far from templated reasoning. RaR-Implicit achieves 31% relative improvement here, suggesting strong applicability beyond narrow reasoning tasks.
>
> To recap, our goal in this paper is controlled experimentation as stated in Sec. 9. We focus on medicine and science for specific reasons: HealthBench provides a real-world, rubric-graded conversational evaluation,and GPQA-Diamond lets us demonstrate gains on a commonly adopted challenging RLVR benchmark. This scope is a foundation rather than a claim of universal coverage, and we view RaR as a path toward broader domains.
>
>
> > Aggregation strategies are static; dynamic or learned weighting not explored.
>
>
> We explicitly discuss this in Sec. 9 (Limitations) as a natural extension and advanced weight aggregation strategies: learning continuous weights and dynamic weighting.
>
>
> Our focus here is to establish the core methodological foundation and demonstrate that rubrics, regardless of weighting strategy, provide a robust supervision signal. A few clarifications:
> - The goal of RaR-Explicit is interpretability under controlled settings; hence weights are intentionally simple and not optimized. Despite simple weights, RaR-Explicit is still the second-best method overall.
> - RaR-Implicit (our best method) demonstrates that learned aggregation is possible even without weight tuning.
>
> We hope these clarifications provide the right context for evaluating our contribution.

---

> > ### Author Response · Authors · 2025-11-26
> > **Follow-up with Reviewer 19Mf**
> >
> > Thank you again for your thoughtful and constructive review of our paper. We have submitted our rebuttal and updated the draft, and we hope our responses address the weaknesses you raised. As the discussion period is ending soon, we would really appreciate it if you could take a quick look at our replies and let us know if any issues remain.

---

> > ### Comment · Reviewer_19Mf · 2025-11-27
> > **Response from Reviewer**
> >
> > Thanks for your response. I will maintain my score.

---

### Official Review · Reviewer_xETQ · 2025-10-31

**Soundness:** 3
**Presentation:** 2
**Contribution:** 3
**Rating:** 4
**Confidence:** 5

**Summary:**

This paper introduces Rubrics as Rewards (RaR), an RL framework that uses rubric-based feedback as structured reward signals for policy training, extending RL beyond verifiable conditions like answer-only rewards. The authors construct instance-specific rubrics in medical and scientific fields and evaluate multiple strategies for translating rubric feedback into rewards. Experiments show that RaR achieves 31% and 7% relative improvements on HealthBench and GPQA-Diamond, respectively, and that rubric-based rewards enable more consistent alignment with human preferences as model scale increases.

**Strengths:**

1. The authors propose an innovative approach that uses rubric-based rewards to train models on tasks lacking verifiable reward signals in RL.
2. The experiments and analyses demonstrate that rubric-based rewards align well with expert human preferences and are effective when the LLM-as-a-Judge model is sufficiently capable.
3. The authors conduct experiments in the medical and scientific domains to demonstrate the effectiveness of the proposed rubric-based reinforcement learning paradigm.

**Weaknesses:**

1. The authors employ large-scale, instance-level rubric labeling for all training samples and use an LLM-as-a-Judge model (GPT-4o-mini) to generate rubric-based rewards in training. While this approach incurs substantial computational costs—since each rubric evaluation consumes additional tokens—the resulting performance gains, particularly on GPQA, are marginal (only 1.1 points compared to the Reference-Likert baseline).

2. This paper uses only a single model, Qwen2.5-7B, as the policy model. However, experiments on a single model cannot fully demonstrate the robustness of the proposed method. Since the labeled rubrics are policy-agnostic, incorporating them into RL training for other policy models would be straightforward and could better show the method’s comprehensiveness and robustness.

3. The presentation of the paper could be improved — for example, Sections 4.1 to 4.5 contain many experimental settings that could be presented more intuitively. Using tables to summarize these details would significantly enhance the paper’s readability.

**Questions:**

1. How are the final checkpoints selected for evaluation in the RL-based baselines and RaR? Are they the final checkpoints, or are they chosen based on a validation set? If the checkpoints are selected based on intermediate validations, provide the intermediate validation plots could help clarify the effectiveness of the method.

---

> ### Author Response · Authors · 2025-11-21
> **Response to Reviewer xETQ**
>
> We thank the reviewer for recognizing the method’s novelty and its strong alignment with human preference. We have incorporated the suggested improvements and address the weaknesses and questions below:
>
> > Weakness 1: Rubric evaluation approach incurs substantial computational cost
>
> This is not true, our method does *not* incur substantial computational cost as compared to the baselines. In our setup, RaR-Implicit and Reference-Likert both use a single judge call per sample and have nearly identical output token lengths, since both ask the judge model to produce a single scalar JSON score (Appendix A.7).
>
> The minor additional cost comes mainly from the rubric text included in the input, not from generating substantially more tokens or making extra LLM calls. We did a quick analysis of token lengths and found that adding the rubric increases the judge prompt by only 94 and 117 tokens on average in RaR-Medicine and RaR-Science respectively, which is a modest overhead. Moreover, RaR-Explicit also uses only a *single judge call* per prompt (not multiple calls), with a compact per-criterion dictionary output which is only a minor addition to output token length.
>
> > Weakness 2: Robustness across policy models.
>
> Thank you for this suggestion to test the robustness of our method. To wit, we trained a smaller policy model, \texttt{Qwen2.5-3B}, using the same three reward strategies. As shown in Table 5, the outcome remains consistent: RaR-Implicit improves over Direct-Likert by +7.8 points and over Reference-Likert by +3.6 points on HealthBench-1k. This provides a clear additional validation point to illustrate that rubric-based reward signals can be applied effectively to a different and more constrained policy model.
>
> | Baseline/Method          | Overall Score |
> |--------------------------|--------------:|
> | Qwen2.5-3B (Base Policy) | 4.13%         |
> | Direct-Likert            | 13.74%        |
> | Reference-Likert         | 17.95%        |
> | RaR-Implicit (ours)      | **21.55%**    |
>
>
> > Weakness 3: Presentation improvements.
>
> We appreciate the reviewer’s suggestion about improving presentation clarity in Sections 4.1 to 4.5. In response, we have added Table 1, which summarizes all baseline methods, their training settings, aggregation types, and reward grounding. This improves readability and allows quick comparison across methods.
>
>
> > Q1: Checkpoint Selection
>
> All trained models for main results use the final checkpoint at 300 training steps (Table 10). In our runs, all reward strategies show stable convergence by this point, making the final-checkpoint evaluation representative of each method’s performance.

---

> > ### Author Response · Authors · 2025-11-26
> > **Follow-up with Reviewer xETQ**
> >
> > Thank you for your detailed review and helpful suggestions. We have carefully addressed all of your comments in the rebuttal and integrated your feedback into the revised version of the paper, with modifications shown in red. We would be happy to provide any further clarification. Since the discussion deadline is near, we would greatly appreciate it if you could take a moment to look over our responses.

---

> > > ### Comment · Reviewer_xETQ · 2025-11-27
> > >
> > > Thanks for the clarification. I raised my score

---

### Official Review · Reviewer_1uK3 · 2025-11-01

**Soundness:** 3
**Presentation:** 2
**Contribution:** 2
**Rating:** 4
**Confidence:** 3

**Summary:**

This paper proposes Rubrics as Rewards (RaR), a reinforcement learning framework that extends RL with Verifiable Rewards (RLVR) to domains lacking clear correctness signals—such as medical and scientific reasoning—by leveraging structured, instance-specific rubric criteria as reward functions. The authors empirically evaluate strategies for aggregating rubric feedback and demonstrate strong improvements over conventional LLM-as-judge Likert-style baselines on HealthBench (medicine) and GPQA-Diamond (science) datasets. The paper details rubric construction, aggregation schemes, experimental outcomes, and ablation studies, highlighting the benefits of rubrics for judge consistency and alignment with human preferences.

**Strengths:**

1 The paper tackles a timely and important limitation of RLVR by moving beyond domains with verifiable, binary reward signals, introducing a pragmatic and interpretable alternative via structured rubrics for open-ended reasoning.

2 Empirical evaluation is thorough, utilizing large and diverse medical and science benchmarks (HealthBench, GPQA-Diamond), and benchmarking several methods including Direct-Likert, Reference-Likert, and filtered SFT, ensuring a fair assessment.

3 The paper offers the RaR-Medicine and RaR-Science datasets with rubric annotations, valuable resources for the community.

**Weaknesses:**

1 Limited Generalization and Scope: Experimental evaluation is confined to domains with relatively structured reasoning (medicine, science) and primarily question-answering tasks. The approach is not shown to generalize to less-structured or dialog/agentic settings noted in Section 9.

2 Implementation Ambiguities: Category-to-weight mapping for explicit aggregation ("Essential": 1.0, etc.) is somewhat arbitrary, and no optimization or sensitivity analysis is provided. Table 2 suggests some robustness, but further technical depth would strengthen trust and generalizability.

3 Mathematical Ambiguities: The extension from binary to continuous $c_j(x, \hat{y})$ (as briefly suggested) is not developed; implications for policy learning, reward distributions, and judge reliability are not analyzed.

**Questions:**

1 For the implicit aggregation strategy (Eq. 2, Page 3), can the authors provide a deeper technical or empirical analysis of $f_\phi$ as an LLM judge? Specifically, how sensitive is this function to prompt design, rubric length, or underlying model drift?

2 Can the authors discuss the feasibility of extending rubric design and aggregation to dialog, tool-use, or more open-ended agentic tasks, and are there preliminary results or design recommendations for such settings?

3 Clarify how continuous (non-binary) rubric criteria (as alluded to on Page 3) would alter reward distributions and policy learning dynamics. Are there settings (Medicine/GPQA) where continuous-valued scores were tried, and if so, what were the findings?

---

> ### Author Response · Authors · 2025-11-21
> **Response to Reviewer 1uK3**
>
> We thank the reviewer for their thoughtful comments and finding our work **timely** to address important limitations of RLVR, our evaluation to be **thorough** capturing diverse domains, and including **compelling baselines** for fair assessment.
>
> Before addressing each point, we note that most of the weaknesses that were pointed out are *already stated* in the Limitations and Future Work section, which outlines the broader extensions of our work beyond the scope of this paper. We provide our detailed response below:
>
>
> - ### Limited Generalization and Scope:
> > W1: approach is not shown to generalize to less-structured or dialog/agentic settings noted in Section 9.
>
>
> We respectfully disagree that our study is confined to “structured reasoning.” HealthBench evaluation set already captures dialog based assessment, since it consists of real-world, multi-turn clinical conversations for seeking medical advice. The output being graded are also free-formed text / conversation responses. We therefore expect our findings to translate to dialog-based settings. Additionally, Section 9 of the paper is our Limitations section, where we explicitly acknowledge that we focus on medicine and science to enable controlled experiments and identify broader validation across dialog, tool-use, and agentic tasks as important future directions. Our intent was to introduce a novel reward mechanism and present a controlled study rather than to claim exhaustive domain coverage.
>
>
> > Q2: feasibility of extending rubric design and aggregation to dialog, tool-use, or more open-ended agentic tasks, preliminary results or design recommendations for such settings
>
> Our current results already show clear signs of life in dialog through HealthBench. Extending rubric-conditioned rewards to tool-use and agentic training is a natural next step, and we outline this in Section 9. Encouragingly, concurrent works are beginning to explore closely related ideas, using rubric or checklist-style rewards to guide multi-step policies in complex instruction-following, tool-use, and agentic settings [1,2].
>
> Overall, we view RaR as a general recipe for structured, instance-specific reward design. While this paper establishes the foundation in two controlled domains, we are already seeing aligned trends emerge elsewhere, suggesting the approach is broadly applicable.
>
> **References:**
> [1] Team, Kimi, et al. "Kimi k2: Open agentic intelligence." arXiv preprint arXiv:2507.20534 (2025).
>
> [2] Y. He, et al. "Rubric-Based Benchmarking and Reinforcement Learning for Advancing LLM Instruction Following" arXiv preprint arXiv:2511.10507 (2025).
>
>
> - ### Weight mapping:
> > W2: Category-to-weight mapping for explicit aggregation ("Essential": 1.0, etc.) is somewhat arbitrary, and no optimization or sensitivity analysis is provided.
>
> The category-to-weight mapping in RaR-Explicit is kept simple by design. The scope of this work is to introduce rubrics as rewards and study them under controlled conditions, not to optimize weighting schemes. As already stated in the paper, RaR-Implicit yields the strongest overall results, while RaR-Explicit uses fixed weighted sums to offer more control and interpretability and is consistently competitive, typically ranking just behind RaR-Implicit across our evaluations. We view the choice between implicit and explicit aggregation as application-dependent and leave it to practitioners. We also explicitly note in the scope and Limitations that learned or dynamic weighting is a natural next step.
>
> - ### Mathematical Ambiguities: binary → continuous
> > W3: The extension from binary to continuous (as briefly suggested) is not developed’
> > Q3 (continuous criteria implications / tried?):
>
> Clarification: Our paper clearly states that all rubric criteria scores are binary in every experiment: c_j(x,\hat{y}) \in \{0,1\} (Lines 103–104). The brief mention of continuous criteria appears only to note that Eq. (1) is mathematically generalizable; we do not implement, tune, or evaluate continuous rewards in this work. Developing a continuous per-rubric scoring scheme is orthogonal to this paper.
> Regarding continuous rubric scores, we believe they could yield denser reward signals and potentially lower variance, but would require careful calibration to control judge drift and align scales across criteria. Investigating these dynamics is a distinct research direction that we leave for future work.

---

> ### Author Response · Authors · 2025-11-21
> **Response to Reviewer 1uK3**
>
> > Q1: sensitivity of RaR-Implicit to prompt design, rubric length, or underlying model drift?
>
> Thank you for the question. From the existing experiments in our paper, we can already draw several robustness insights about RaR-Implicit. We recap our findings:
>
> - *Prompt design:* We keep the Reference-Likert judge prompt intentionally similar to RaR-Implicit, with only minor formatting differences besides adding the rubric block. So the improvement cannot be attributed to prompt engineering; it reflects the effect of rubric conditioning. (Appendix A7)
>
> - *Rubric length:* Our ablation in Table 3 shows richer and comprehensive rubrics help under implicit aggregation: Only using essential rubrics underperforms policies trained with all rubrics (34.9% vs 37.2%), so gains are not from a minimal or cherry-picked checklist.
>
> - *Model drift / judge choice:* The implicit rubric advantage holds across judge sizes (Table 4 / Fig. 3), indicating the effect is stable to the underlying judge model and not fragile to drift.

---

> > ### Author Response · Authors · 2025-11-26
> > **Follow-up with Reviewer 1uK3**
> >
> > Thank you again for your detailed review and comments. We have addressed your points in our responses, and as the discussion deadline is approaching, we would appreciate it if you could briefly review them and let us know if any issues remain.

---

### Author Response · Authors · 2025-12-03
**Rebuttal Summary**

We highlight the key strengths and contributions of our work as follows:

* **Pioneering RLVR for non-verifiable domains:** Reviewers recognized our work as a timely extension of the RLVR framework that moves beyond binary correctness to real-world reasoning tasks (reviewers *1uK3*, *xETQ*, *19Mf*).
* **Strong empirical performance:** Reviewers commended the significant improvements on HealthBench and GPQA-Diamond with clear gains over strong baselines (reviewers *1uK3*, *xETQ*, *19Mf*, *dV3e*).
* **Valuable resources:** The release of the RaR-Medicine and RaR-Science datasets with rubric annotations was highlighted as a major contribution (reviewers *1uK3*, *19Mf*).
* **Alignment with human preferences:** Reviewers noted that our rewards successfully align policies with expert human preferences (reviewers *xETQ*, *dV3e*).

We also summarized reviewers’ major concerns and how we addressed them as follows:

* **Generalization and Scope** (reviewers *1uK3*, *xETQ*, *19Mf*): Reviewers asked about the generalization of our method to other domains or evaluation styles. We clarified that our results cover both less structured/dialogue settings (HealthBench) and multiple-choice styled questions (GPQA-Diamond). This demonstrates that RaR generalizes effectively beyond rubric-based evaluation and does not overfit to a specific evaluation style. Specifically, *1uK3* stated that our approach "is not shown to generalize to less-structured or dialog/agentic settings." We clarified that this stems from a misunderstanding of HealthBench, which is a less-structured, free-form dialog task. We also clarified that we did not train on HealthBench, proving our method generalizes effectively without overfitting to specific evaluation styles.
  * ***xETQ*** **raised score**
  * ***19Mf*** **kept a positive score**
  * ***1uK3*** **did not respond**
* **Marginal Gains vs. Baselines** (reviewers *xETQ*, *dV3e*): Reviewers noted overlapping confidence intervals in earlier plots. We conducted new experiments with a larger sample size (N). These results show the overlap was strictly due to small N and statistically confirm the performance gains.
  * ***xETQ*** **raised score**
  * ***dV3e*** **raised score**
* **Computational Cost** (reviewers *xETQ*, *19Mf*): Concerns were raised about the inference cost of using an LLM-as-judge during training. We clarified that our method does not incur substantial computational cost compared to baselines. Our additional analysis confirms the extra cost from rubric text is minor (only \~100 tokens on average).
  * ***xETQ*** **raised score**
  * ***19Mf*** **kept a positive score**

Lastly, we are encouraged that Reviewers *xETQ* and *dV3e* have increased their scores and Reviewer *19Mf* has maintained a positive assessment.

**We believe the remaining reviewer (*1uK3*) would agree with our rebuttal for the following reasons:**

* **Generalization:** Their primary concern regarding generalization was based on the factually incorrect premise that HealthBench is not a dialog setting; our clarification that HealthBench is an unstructured dialog task directly resolves this.
* **Minor Concerns:** We have addressed their remaining questions regarding weight mapping, formulation, and prompt sensitivity through improved presentation and the additional sensitivity analysis included in our rebuttal.

We hope the AC will consider these updates and the strong consensus among the other three reviewers in the final assessment.

---

### Meta-Review · Area_Chair_GZLA · 2026-01-08

**Summary:**

This paper proposes “Rubrics as Rewards,” using instance-specific rubric checklists to provide more structured reward signals for RL post-training beyond verifiable tasks. The approach is clearly described (explicit vs implicit rubric aggregation) and supported by a broad empirical evaluation on HealthBench and GPQA, with relevant baselines (Direct-Likert and Reference-Likert) under a consistent GRPO setup. The paper also includes useful analyses, including judge-scale alignment improvements and ablations demonstrating that rubric quality and construction matter, which strengthens the claim that gains come from rubric conditioning rather than superficial prompt changes.

Reviewer feedback raised reasonable questions about generalization beyond the studied domains, the meaningfulness of improvements under variance, and computational overhead. The authors’ clarifications and additional evaluations (including higher-powered GPQA runs) address several of these concerns, though the work remains primarily an empirical recipe rather than a fundamentally new RL method. Overall, I view this as a solid and timely contribution to reward design for RLHF-style training in non-verifiable domains, and recommend acceptance.

**Reviewer Concerns:**

Reviewer 1uK3. The rebuttal clarified several of 1uK3’s points: it corrected a misunderstanding about HealthBench (that the setting is not meant to be a multi-turn dialog/agent benchmark), explained the scope of the work as reward design for expert QA in medicine/science rather than general tool-use agents, and made the positioning relative to RLVR and existing RLHF methods more explicit. However, 1uK3’s broader concern—that the method’s impact may be narrow and that the paper does not fully demonstrate benefits beyond the specific rubric-based datasets—remains only partially addressed.

Reviewer dV3e. The authors directly addressed dV3e’s main factual concerns about training and fairness of baselines, clarifying that all methods in Figure 2 are trained on the same RaR-Medical data and do not use HealthBench rubrics at training time, and providing additional evidence (including a 100-run GPQA evaluation and HealthBench training ablation) to show that the gains over Reference-Likert are statistically meaningful. dV3e acknowledged these clarifications, admitted mistakes in the initial review, and raised their rating. Some of their more general requests for additional datasets and baselines remain open as future work, but the key misunderstandings were resolved.

Reviewer xETQ. For xETQ, the rebuttal responded to concerns about computational cost, checkpoint selection, and robustness across policy models by quantifying the extra rubric overhead relative to Reference-Likert, explaining the validation and checkpointing strategy, and adding experiments with a smaller policy model (Qwen2.5-3B). These clarifications address most of the practical questions. What remains to some extent is the high-level concern about the method’s cost–benefit tradeoff and its generality beyond the evaluated domains, which the paper acknowledges as a limitation.

Reviewer 19Mf. The authors engaged with 19Mf’s comments by clarifying the formal relationship between rubrics-as-rewards and RLVR, elaborating on rubric generation and weighting choices, and adding more discussion and ablations around rubric design. While these responses help with understanding and partially strengthen the methodological story, 19Mf ultimately chose to maintain their score, suggesting that residual skepticism about the breadth of the contribution and the need for more extensive validation remains.

**Reviewer Scores:**

xETQ and dV3e explicitly say they raised their scores; 19Mf explicitly says that he keeps theirs; 1uK3 is silent after rebuttal.

---

### Decision · Program_Chairs · 2026-01-26

Accept (Poster)